# Acute Effects of Sprint Interval Training and Chronic Effects of Polarized Training (Sprint Interval Training, High Intensity Interval Training, and Endurance Training) on Choice Reaction Time in Mountain Bike Cyclists

**DOI:** 10.3390/ijerph192214954

**Published:** 2022-11-13

**Authors:** Paulina Hebisz, Cristina Cortis, Rafal Hebisz

**Affiliations:** 1Department of Physiology and Biochemistry, Wroclaw University of Health and Sport Sciences, 51-612 Wroclaw, Poland; 2Department of Human Sciences, Society and Health, University of Cassino and Lazio Meridionale, 03043 Cassino, Italy

**Keywords:** choice reaction time, sprint interval training, cycling

## Abstract

This study evaluated the acute effects of sprint interval training and chronic effects of polarized training on choice reaction time in cyclists. Twenty-six mountain bike cyclists participated in the study and were divided into experimental (E) and control (C) groups. The cyclists trained for 9-weeks and performed five training sessions each week. Types of training sessions: (1) sprint interval training (SIT) which consisted of 8–16, 30 s repetitions at maximal intensity, (2) high-intensity interval training (HIIT) included 5 to 7, 5-min efforts at an intensity of 85–95% maximal aerobic power (Pmax), and (3) endurance training (ET) performed at an intensity of 55–60% Pmax, lasting 120–-180 min. In each week the cyclists performed: in group E a polarized training program, which included 2 × SIT, 1 × HIIT and 2 × ET, while in group C 2 × HIIT and 3 × ET. Before (acute effects) and after the 9-week training period (chronic effects) participants performed laboratory sprint interval testing protocol (SITP), which consisted of 12 maximal repetitions lasting 30 s. During SITP maximal and mean anaerobic power, as well as lactate ion concentration and blood pH were measured. Choice reaction time (RT) was measured 4-times: before and immediately after the SITP test—before and after the 9-week training period. Evaluated the average choice RT, minimal choice RT (shortest reaction), maximal choice RT (longest reaction), and the number of incorrect reactions. Before the training period as acute effects of SITP, it was observed: a shorter average choice RT (F = 13.61; *p* = 0.001; η^2^ = 0.362) and maximal choice RT (F = 4.71; *p* = 0.040; η^2^ = 0.164), and a decrease the number of incorrect reactions (F = 53.72; *p* = 0.000; η^2^ = 0.691), for E and C groups. After the 9-week training period, chronic effects showed that choice RT did not change in any of the cyclists’ groups. Only in the E group after the polarized training period, the number of incorrect reactions decreased (F = 49.03; *p* = 0.000; η^2^ = 0.671), average anaerobic power increased (F = 8.70; *p* = 0.007; η^2^ = 0.274) and blood pH decreased (F = 27.20; *p* = 0.000; η^2^ = 0.531), compared to the value before the training period. In conclusion, a shorter choice RT and a decrease in the number of incorrect reactions as acute effects of SITP, and a decrease in the number of incorrect reactions and higher average power as chronic effects of the polarized training program are beneficial for mountain bike cyclists.

## 1. Introduction

Reaction time (RT) can be defined as the time that elapses from the appearance of the stimulus to the response and is considered a good measure for assessing the cognitive system’s ability to process information [1,2,3]. Physiologically, RT is a complex phenomenon that depends on the speed of the sensorimotor cycle, which consists of the detection of the initial stimulus, the transmission of information through the afferent nerves, the generation of a response from the central nervous system, and the final response [4,5]. A distinction is made between simple reaction time, complex reaction time, and choice reaction time [6]. Simple reaction time requires the participant to respond to the presence of a single stimulus. Complex reaction times require the participant to respond to one specific stimulus and to withhold a response when other types of stimuli are presented. Choice reaction times require separate responses for each stimulus type [6]. As athletes need to make decisions quickly and accurately during intense exercise, RT, among other indicators, has been evaluated in different sports [7,8,9,10].

One sport that requires a high-intensity effort during competition is undoubtedly mountain bike racing. During a race, it difficult is not to fall, damage the bike, or collide with another cyclist, which often derails the chances of getting a good result. This is due to the characteristics of the discipline, in which competition takes place on narrow forest paths with uneven ground, sharp turns, and obstacles in the form of stones and roots. In addition, races take place in shifting weather conditions and in direct contact with other cyclists whose behavior cannot be predicted [9,11,12,13]. Therefore, competing in mountain bike racing, in addition to a high level of aerobic and anaerobic capacity [14,15], requires specific technical abilities related to bike control, balance skills [16], good concentration, and short RT [14].

During mountain bike races, high-intensity activity is repeatedly performed, with power output exceeding maximal aerobic power [17]. It has been shown that power output and oxygen uptake measured in sprint interval tests (involving repeated sprints, such as the Wingate test) correlate with race performance [15,18,19,20,21]. The effort exerted in such tests causes a large acid-base imbalance [19,21] highlighting the development of fatigue [22]. Whereas fatigue can cause longer RT among athletes of different sports [7,23]. Numerous studies have assessed the reaction time after a single training session as an acute effect [7,8,24]. For example, Pavelka et al. [24] observed that simple RT deteriorated among MMA fighters after just a single bout of high-intensity exercise such as the Wingate test. On the other hand, Delignières et al. [7] reported that choice RT improved during an intense 4-min effort among fencers. Kashihara and Nakahara [8] also observed an improvement in choice RT during intense exercise performed at constant power among healthy male students. The above studies cover different exercise protocols and different groups of athletes or non-training participants [7,8,24]. The maximal 30-s effort performed in the study of Pavelka et al. [24] caused RT deterioration, while in the studies of Delignières et al. [7] and Kashihara and Nakahara [8], RT improved, but the maximal intensity was not used there, only a 4-min exercise at an intensity of 80% maximal aerobic power [7] and a 10-min exercise at lactate threshold [8]. Based on the available literature, it is difficult to clearly determine the acute effects of sprint interval training (consisting of repeated maximal efforts) on the choice RT in a group of mountain bike cyclists who regularly perform high-intensity exercise.

In endurance sports, including mountain bike cycling, training is increasingly intensified through the use of a polarized training strategy, which consists of sustained low-intensity training as well as high-intensity interval training [25,26,27]. The volume of low-intensity training sessions is approximately 80% of the total training volume, while high-intensity training is approximately 20% of the total training volume [28]. In polarized training programs, moderate-intensity training at the level of the second ventilatory threshold (VT2) is not used [25,29,30], or these training sessions account for a small part of the training load (up to 5% of the total training volume) [31]. In the studies by Hebisz et al. [29,30], a polarized training program was defined as training including sprint interval training (SIT), high-intensity interval training (HIIT), and low-intensity endurance training (LIT), with the exclusion of training at an intensity of VT2. These studies [29,30] have shown that the 8–9 weeks of a polarized training program is effective in improving aerobic capacity assessed by maximal oxygen uptake (VO_2_max) and maximal aerobic power, among mountain bike cyclists. The available literature lacks information on the effect of a polarized training program (including sprint interval training, high-intensity interval training, and endurance training) on choice reaction time. Interesting is not only the acute effects of single highly intense training sessions but also the long-term effects described as chronic effects because athletes perform training systematically and over a long period of time. It has been shown that moderate-intensity training or high-intensity interval training performed for several weeks improves reaction time, among untrained participants [32,33]. Sherwood and Selder [34] observed that endurance athletes have a shorter reaction time compared to the untrained population. According to Reigal et al. [35], the level of physical activity and physical capacity of the participants correlated with reaction time. It is believed that high levels of physical activity improve cognitive function. Since systematic exercise affects molecular mechanisms, including an increase in the concentrations of brain-derived neurotrophic factor (BDNF) and insulin-like growth factor-1 (IGF-1) in the brain and blood, which are involved in the process of neuro-creations [36]. In previous studies [29], it was shown that a polarized training program in cyclists reduces the serum BDNF concentrations during and after the sprint interval test, as acute effects. Similar observations were reported by Nofuji et al. [37], that participants with high physical activity may be able to utilize BDNF rapidly as an acute effect of high-intensity exercise. If polarized training among athletes causes changes in BDNF concentration, it can be assumed that the use of polarized training affects the process of neurons and synapse formation or differentiation, as described by Bos et al. [38]. The above-mentioned changes in the nervous system may contribute to the improvement of cognitive function [39]. One of the measures of cognitive function is reaction time [40]. Therefore, it can be assumed that polarized training may induce changes in RT. Regardless of the above reports, it has been shown that the polarized training program significantly improves cardiac output [21]. High cardiac output determines blood flow, and cerebral blood flow is negatively correlated with reaction time [41]. Moreover, the literature suggests a significant influence of lactate concentration on cognitive function [42]. The increase in lactate concentration is characteristic of the SIT [21] and HIIT training [42], which are part of a polarized program. Lactate ions can cross the blood-brain barrier and affect the expression and release of BDNF [43]. The above reports support the assumption that a polarized training program may positively affect cognitive function, including reaction time.

Therefore, the aim of this study was to determine the acute effects of sprint interval training and chronic effects of 9-week polarized training (which included sprint interval training, high-intensity interval training, and endurance training) on choice RT in mountain bike cyclists.

It was assumed that the acute effect of a single SIT training session including maximal effort would be longer choice RT, while the chronic effect of a 9-week polarized training program would be shorter choice RT, in group E.

## 2. Materials and Methods

### 2.1. Participants

Twenty-six mountain bike cyclists participated in the study. Each participant was characterized by at least three years of training experience in cycling. The participants were randomly divided into two groups: experimental (E, *n* = 14, including ten men and four women) and control (C, *n* = 12, including nine men and three women). The characteristics of the groups are shown in Table 1. Additionally, the characteristics of men and women included in each of the studied groups are presented (Table 1).

The study design was approved by the Ethics Committee of the Wroclaw University of Health and Sport Sciences and carried out in accordance with the Declaration of Helsinki. Written informed consent was obtained from the participants and their guardians after the study details, procedures, and benefits and risks were explained.

### 2.2. Test Procedures

Prior to the experiment, participants completed an incremental test (IT) to characterize the study groups. Laboratory tests were performed immediately before and after the experiment, which included performing a sprint interval testing protocol (SITP) and measuring choice reaction time. During the 24 h that preceded the exercise tests, the participants did not do any training. There was a break of 48 h between IT and SITP. All the tests were carried out under controlled laboratory conditions (temperature and air humidity) in the Exercise Laboratory of the Wroclaw University of Health and Sport Sciences (certificate PN-EN ISO 9001:2001). The same two researchers performed the measurement both before and after the experiment in all participants, and the researchers were blind to the research question.

#### 2.2.1. Incremental Test (IT)

The test was conducted on a Lode Excalibur Sport cycle-ergometer (Lode BV, Groningen, the Netherlands), calibrated before the study. The effort started with a load of 40 (women) or 50 W (men), every 3 min the load was increased by 40 (women) or 50 W (men) until the participant refused to continue. If a participant was unable to complete an entire 3 min stage 0.22 (women) or 0.28 W (men) per second missed was subtracted from the work rate at that stage [20,30]. The highest power output determined in the incremental test was taken to be the measure of maximal aerobic power (Pmax), as described by Peyré-Tartaruga and Coertjens [44]. The Pmax value was used to individually determine the intensity during the training performed in the experiment.

Respiratory function was measured during the test. The cyclist wore a mask connected with a Quark gas analyzer (Cosmed, Rome, Italy). The gas analyzer was calibrated before use with a reference gas mixture of carbon dioxide—5%, oxygen—16%, and nitrogen—79%. Respiratory parameters were measured in each recorded breath (breath-by-breath) and then averaged over 30-s intervals. Based on the recorded data, the maximal oxygen uptake (VO_2_max) was determined, as the highest 30-s average. As a confirmation of the maximal value, the plateau phase of VO_2_max was determined according to the method proposed by Edvardsen et al. [45], which was modified for the incremental test consisting of 3-min steps. Oxygen uptake (VO_2_) values were compared between all 30 s intervals of the last step and the one 30 s interval of the penultimate step with the highest VO_2_. A plateau phase of VO_2_max was considered to occur when VO_2_ did not differ by more than 1.5 mL·kg^−1^·min^−1^ [46,47] between the indicated 30 s intervals. Participants who did not reach the plateau phase of VO_2_max were not included in the data analysis. Therefore, in the presented study, 26 cyclists out of 30 who started the study were included in the data analysis and description of the results.

#### 2.2.2. Sprint Interval Testing Protocol (SITP)

The test was also conducted on a Lode Excalibur Sport cycle-ergometer. The test was preceded by a 20-min warm-up at an intensity of 40% Pmax for 5 min and then at an intensity of 60% Pmax for 15 min. The warm-up was followed by a low-intensity active rest of 10 min. Then, the participant performed 12 maximal repetitions lasting 30 s, during which the participant was to obtain the highest possible power in the shortest possible time and maintain it for as long as possible. Repetitions were divided into 3 sets and 4 repetitions were performed in each set. Between repetitions, a low-intensity (with power below 50 W) active rest of 90 s was used. A 25-min active rest was used between sets, during which the first 2 min were performed at an intensity of 20% Pmax, the next 20 min at an intensity of approx. 50% Pmax, and the last 3 min at an intensity of 20% Pmax. The duration and intensity of the active rest between sets were selected based on previous research [19]. The course of the sprint interval testing protocol was similar to the sprint interval training that the cyclists performed during the experiment.

Power was measured during each repetition. In the data analysis, peak power (Ppeak) and average power (Pav) measured during all repetitions performed was used. During the test, heart rate (HR) was recorded using a V800 heart rate monitor (Polar, Oy, Finland). 

Immediately before the test and in the third minute after the end of each set arterialized capillary blood was drawn, to determine pH and lactate ion (La^−^) concentrations using the RAPIDLab 348 (Siemens Healthcare, Germany) and Lactate Scout (SensLab, Leipzig, Germany) analyzers, respectively.

#### 2.2.3. Choice Reaction Time Measurement

The choice RT was measured using an MCZR/ATB 1.0 m (ATB INFO-ELEKTRO, Zabrze, Poland), validated by Zapała et al. [48]. Measurements were taken immediately before and immediately after the sprint interval testing protocol (after the third/last sets). The reaction time meter consisted of a central control panel connected to a beacon for visual and auditory stimuli together with a stimulus reception panel. The stimulus beacon emits visual stimuli in three colors: red, orange, green, and auditory stimuli in two sounds: high (treble) and low (bass). The stimulus reception panel consisted of two hand-held buttons (one right and one left) placed on cables terminated with a JACK-type plug. During the measurement, the participant sat on the cycle-ergometer with hands on the handlebars and simultaneously held push-button plugs in their hands. The participant was positioned facing the stimulus beacon at a distance of 3 m. Figure 1 shows the position that the cyclists took during the reaction time measurement. The participants were asked to react as quickly as possible by pressing a button with their right hand after a red light appeared and with their left hand after a green light appeared. Intentionally, the button in the right hand was red, while the one in the left hand was green. The participants did not have to react to the remaining stimuli: orange light and sounds. Each cyclist had a test measurement for familiarization, which lasted 40 s and consisted of 10 stimuli, and was performed only once (on the day the cyclists performed an incremental test). In contrast, the main measurement lasted 120 s and included 20 stimuli arranged in a specific order and occurring at a specific time. The choice RT measurement was applied to all participants, before and immediately after the sprint interval testing protocol and before and after the experiment. The measurement evaluated the average choice RT (time calculated from all reactions), the minimal choice RT (shortest reaction), the maximal choice RT (longest reaction), and the number of incorrect reactions (no reaction when supposed to react, reaction when not supposed to react, reaction with the wrong button, reaction between stimuli).

### 2.3. Experiment Schedule

Prior to the experiment, for six weeks, each participant limited the volume of training to three sessions per week, in which the intensity did not exceed 70% of HRmax. The experiment lasted nine weeks and was conducted during the preparatory season. The types of training sessions performed during the experiment are shown in Table 2. During the experiment, group E athletes performed:-sprint interval training (SIT), twice a week. The training was preceded by a 20-min warm-up at an intensity of 40% Pmax for 5 min and then at an intensity of 60% Pmax for 15 min. The warm-up was followed by a low-intensity active rest of 10 min. SIT training consisted of 8–16 repetitions at maximal intensity, lasting 30 s. (During the 1st–3rd week of the experiment the cyclists performed eight repetitions, during the 4th–6th week—12 repetitions, in the 7th–9th week—16 repetitions). Efforts were divided into sets and four repetitions were performed in each set. Between repetitions, a low-intensity (with power below 50 W) active rest of 90 s was applied. A 25-min active rest was applied between sets, during which the first 2 min were performed at an intensity of 20% Pmax, followed by 20 min at an intensity of approx. 50% Pmax, and for the last 3 min an intensity of 20% Pmax. The course of the SIT was planned analogously to the SITP.-high-intensity interval training (HIIT), once a week. The training was preceded by a 20-min warm-up at an intensity of 40% Pmax for 5 min and then at an intensity of 60% Pmax for 15 min. The warm-up was followed by a low-intensity active rest of 10 min. HIIT training included 5 to 7, 5-min efforts at an intensity of 85–95% Pmax, interspersed with a 12-min workout at an intensity of 50% Pmax. During the 1st–3rd week of the experiment, the cyclists performed five efforts during HIIT training, in the 4th–6th week—six efforts, and in the 7th–9th week—seven efforts.-endurance training (ET), twice a week. The training was preceded by a 15-min warm-up at an intensity of 40% Pmax for 5 min and then at an intensity of 55–60% Pmax for 10 min. The warm-up was followed by a low-intensity active rest of 5 min. ET training was performed at an intensity of 55–60% Pmax, lasting 120–180 min. During the 1st–3rd week of the experiment the training lasted 120 min, in the 4th–6th week—150 min, in the 7th–9th week—180 min.

During the experiment, group C athletes performed:--HIIT training twice a week. The training was preceded by a 20-min warm-up at an intensity of 40% Pmax for 5 min and then at an intensity of 60% Pmax for 15 min. The warm-up was followed by a low-intensity active rest of 10 min. HIIT training included 5 to 7, 5-min efforts at an intensity of 85–-95% Pmax, interspersed with a 12-min workout at an intensity of 50% Pmax. During the 1st–3rd week of the experiment, the cyclists performed five efforts during HIIT training, in the 4th–6th week—six efforts in the 7th–9th week—seven efforts.--ET training three times a week. The training was preceded by a 15-min warm-up at an intensity of 40% Pmax for 5 min and then at an intensity of 55–60% Pmax for 10 min. The warm-up was followed by a low-intensity active rest of 5 min. ET training was performed at an intensity of 55–60% Pmax, lasting 120–180 min. During the 1st–3rd week of the experiment the training lasted 120 min, in the 4th–6th week—150 min, in the 7th–9th week—180 min.

In each group, two days a week were designated for active or passive rest. The total weekly training volume was 10–13 h for each participant in the experiment and did not differ significantly between groups. In the fifth week of the experiment, a recovery cycle was applied which was characterized by a 50% lower total training volume in both groups (5–6.5 h), with no change in training methods or intensity. 

During training, power output was monitored using the PowerTap G3 ANT+ and GS ANT+ system (PowerTap, Madison, WI, USA) and heart rate was monitored using the Garmin Edge 520 and Edge 810 system (Garmin Ltd., Olathe, KS, USA).

### 2.4. Statistical Analysis

Statistica 13.1 software (StatSoft Inc., Tulsa, OK, USA) was used for statistical calculations. The arithmetic mean and standard deviation were calculated for all variables. The Kolmogorov-Smirnov test was used to check whether the values of the analyzed parameters were statistically significantly different from the normal distribution. Analysis of variance with repeated measurements and the Scheffe post-hoc test were used to identify statistically significant differences in reaction time parameters measured before (baseline measurements) and after (exercise measurements) the sprint interval testing protocol, and statistically significant differences in the parameters assessed between groups E and C and between tests performed before and after the experiment. The effect sizes were determined using the eta squared (η^2^). A level of *p* < 0.05 was considered statistically significant for all analyses.

Prior to the experiment, using the G-Power 3.1.9.4 software (StatSoft Inc., Tulsa, OK, USA), we estimated that the minimum total sample size for ANOVA with repeated measurements is 16 people, assuming that we expect a strong effect size, i.e., a partial η^2^ ≥ 0.14 at *p* < 0.05 [49].

## 3. Results

The presented study originally included 30 cyclists, but four cyclists did not reach the plateau phase of VO_2_max, which was the criterion for inclusion in the data analysis. So the results presented in this section include 26 cyclists. All tested parameters were characterized by no significant differences compared to the normal distribution.

### 3.1. Acute Effects

Prior to the experiment, the acute effects of a single dose of sprint interval testing protocol were assessed using ANOVA with repeated measures test. It demonstrated the statistically significant main effect of repeated measurements for average choice RT (F = 13.61; *p* = 0.001; η^2^ = 0.362), maximal choice RT (F = 4.71; *p* = 0.040; η^2^ = 0.164), and the number of incorrect reactions (F = 53.72; *p* = 0.000; η^2^ = 0.691) (Table 3). Based on post-hoc tests, it was shown that after the sprint interval testing protocol, there were statistically significant shorter average choice RT in groups E and C, shorter maximal choice RT in group C, reduction in the number of incorrect reactions in groups E and C, compared to the baseline value measured before the sprint interval testing protocol (Table 3).

### 3.2. Chronic Effects

Using analysis of variance, for baseline measurements taken before a sprint interval testing protocol, a statistically significant main effects group x repeated measures were observed for average choice RT (F = 5.34; *p* = 0.030; η^2^ = 0.182), for the number of incorrect reactions (F = 10.49; *p* = 0.003; η^2^ = 0.304), and the main effect of repeated measures on the number of incorrect reactions (F = 54.21; *p* = 0.000; η^2^ = 0.693). Based on post-hoc tests, it was shown that there was a statistically significant reduction in the number of incorrect reactions in the experimental group after the applied experiment (Table 3).

Using analysis of variance, for exercise measurements taken immediately after a sprint interval testing protocol, a statistically significant main effect of the group for the number of incorrect reactions was observed (F = 42.41; *p* = 0.000; η^2^ = 0.639), the main effect of repeated measurements on the number of incorrect reactions (F = 10.21; *p* = 0.004; η^2^ = 0.298), main effects group x repeated measures on the number of incorrect reactions (F = 49.03; *p* = 0.000; η^2^ = 0.671). Based on post-hoc tests, it was shown that after the experiment, there was a statistically significant reduction in the number of incorrect reactions in the experimental group, compared to the control group and compared to the value before the experiment (Table 3).

Using analysis of variance, there was a statistically significant main effect of repeated measurements (F = 8.70; *p* = 0.007; η^2^ = 0.274) for Pav_3_ and main effects group x repeated measures for Pav_1_ (F = 11.45; *p* = 0.002; η^2^ = 0.323), for Pav_2_ (F = 5.30; *p* = 0.030; η^2^ = 0.181) and for Pav_3_ (F = 5.54; *p* = 0.028; η^2^ = 0.194). Based on post-hoc tests, it was shown that Pav_1_ and Pav_3_ increased statistically significantly in the experimental group after the applied experiment compared to the value obtained before the experiment (Table 4).

A statistically significant main effect of repeated measurements emerged for pH_1_ (F = 13.90; *p* = 0.001; η^2^ = 0.367) and main effects group x repeated measures for pH_2_ (F = 7.70; *p* = 0.010; η^2^ = 0.244) and pH_3_ (F = 27.20; *p* = 0.000; η^2^ = 0.531). Based on post-hoc tests, it was shown that pH_1_ increased statistically significantly in the control group after the experiment compared to the value before the experiment. Furthermore, post-hoc analysis showed that after the applied experiment, pH_3_ statistically significantly decreased in the experimental group, while it increased in the control group, compared to the value before the experiment (Table 5).

## 4. Discussion

In the presented study it was determined acute effects of sprint interval training and chronic effects of polarized training (sprint interval training, high-intensity interval training, and endurance training) on choice RT in mountain bike cyclists. Assessment of acute effects began in the first seconds after the last repetition during the SITP. Chronic effects were assessed after nine weeks of simultaneous use: sprint interval training, high-intensity interval training, and endurance training, which formed the polarized training program. In the presented study, as acute effects, it was observed that after a single dose of sprint interval test, the average choice RT and maximal choice RT were statistically significantly shortened and the number of incorrect reactions reduced, compared to baseline values measured immediately before the exercise test. Similarly, Buchholtz and Burgess [50], Delignières et al. [7], and Kashihara and Nakahara [8] observed improvements in choice RT after intense exercise. Buchholtz and Burgess [50] showed that in a group of cyclists, choice RT was significantly shorter after high-intensity cycling exercise compared to low-intensity exercise. Delignières et al. [7] indicate that choice RT improves with increasing exercise intensity among fencers, but among athletes with no experience in decisional sports choice reaction time deteriorated with increasing exercise intensity. They compared 4-min exercise bouts with an intensity of 20, 40, 60, and 80% maximal aerobic power [7]. Similar conclusions were reached by Durand et al. [51] who evaluated choice RT among athletes of different sports disciplines after cycling at intensities of 35, 60, and 90% VO_2_max. Athletes who trained in team sports improved their choice of RT when exercising at high intensities. However, the improvement in reaction times mirrored an increase in errors. In contrast, in a group of gymnasts and athletes, exercise had no effect on either choice reaction time or the number of errors [51]. The literature shows that as exercise intensity increases, athletes in decisional sports are able to improve their performance in reaction time tasks. Mountain bike cycling can also be considered a decisional sport because, during a race, cyclists often have to make quick decisions and react to changing conditions and situations on the competition course. Moreover, this takes place during intense physical activity [52]. In the present study, in addition to the improvement in RT, the number of incorrect reactions decreased, which distinguishes our results from those of Durand et al. [51] and Delignières et al. [7], who showed an increase in errors. The differences in results regarding the effect of exercise on incorrect reactions are difficult to explain. Perhaps this is related to the use of different exercise protocols or this is the result of different sports specializations—thus different types of exercise adaptation.

Pavelka et al. [24] and Gierczuk et al. [10] also studied reaction time among athletes of decisional sports, but they obtained contradictory results, as they showed that reaction time deteriorated after performing intense exercise. Pavelka et al. [24] used a single Wingate test among MMA fighters. In contrast, Gierczuk et al. [10] applied a variable intensity effort—three rounds of wrestling bouts. After each round, a reaction time measurement was performed. It is possible that the results of the presented study and the results obtained by Pavelka et al. [24] and Gierczuk et al. [10] differ also due to the test protocols used. These study protocols vary in the duration of the reaction time measurements and the time between the end of the exercise and the start of the reaction time measurements. It is worth emphasizing that the deleterious effects are often reduced further out from exercise so the appropriate measuring time is important [6,7]. On the other hand, it is possible that the differences between the effects in the studies in question are influenced by the level of aerobic capacity. In the study by Gierczuk et al. [10], the research group was divided into subgroups according to the ranks occupied in sports competitions. In the higher-classified (presumably with higher aerobic capacity) subgroup, the deterioration in reaction time was less than in the lower-classified group. This resulted in a significant difference in average reaction time between the subgroups in question during measurements taken after the second and third rounds of the fight (in the baseline measurement—and in the measurement taken after the first round of the fight the subgroups did not differ significantly statistically). The present study involved a group of well-trained mountain bike cyclists (men and women) with an initial VO_2_max level of approximately 57 mL∙min^−1^∙kg^−1^. This level, according to Tomaszewski et al. [53] characterizes elite athletes. The high aerobic capacity of the cyclists, in the study presented here, may have favored rapid restitution between sprints and sprint sets in the sprint interval testing protocol used, slowing down fatigue processes. Perhaps because of this, choice RT in the presented study did not worsen after the sprint interval test. On the contrary, the average reaction time we measured after the sprint interval test improved. Such a reaction may be beneficial in terms of preparing cyclists to react quickly when overcoming obstacles in mountain bike racing competitions. Therefore, in the future, it would be interesting to investigate how reaction times would change if several sprints, such as 30 s maximal efforts, were performed as part of the warm-up for training or competition.

In the presented study, as a chronic effect, it was not observed any changes in choice RT, measured at rest (pre-test) and immediately after the SITP test, in any of the study groups as a result of the 9-week experiment. However, we observed that after the applied experiment, only in the experimental group did the number of incorrect reactions significantly decrease, both in measurements taken before and immediately after the SITP test. Furthermore, after the applied experiment, the number of incorrect reactions in the experimental group was significantly reduced compared to the control group, in measurements taken immediately after the SITP test. Findings from other authors indicate that regular training improves reaction time, not just error rates [32,54,55,56,57], however, these studies compared trained to untrained subjects and trained subjects showed faster reaction times than untrained subjects. Van de Water et al. [58] compared badminton-specific reaction time in two groups of badminton players that were differentiated by time spent training and ranking. However, their assumption that athletes who do more training and have a higher ranking should achieve better reaction times was not borne out. They showed that there were no statistically significant differences in badminton-specific reaction time between the two groups of badminton players [58]. Therefore, referring to the information contained in the literature and data presented in this manuscript, it can be supposed that training previously untrained people may improve their reaction time. However, among athletes who train regularly, a higher number or volume of training performed does not affect reaction time, as shown by van de Water et al. [58]. The findings of the presented study indicate that intensification of the training process among trained cyclists, through the use of a polarized training program, also has no effect on reaction time. Despite the lack of significant changes in reaction time, the group performing the polarized training program decreased the number of incorrect reactions during the measurement. The available literature lacks information on the impact of training on incorrect reactions in the test measurements performed. However, it has been shown that the incorrect reaction is followed by the adaptation of neurons, which is observed as an increase in the activity of brain areas related to the performed task [59]. There are also suggestions that among athletes, patterns of behavior are better developed, which allows them to react faster after a mistake, compared to untrained participants [60]. This is particularly evident in the case of athletes specializing in competitions requiring open skills [60,61]. It is possible that the intensive training performed by the experimental group in the presented study allowed to improve the effectiveness of behavior patterns after a mistake. If so, it would be significant for mountain bike cyclists as competing also requires open skills.

Van de Water et al. [58] suggest that the elite group of athletes performs better in sports as a result of higher tactical skills and higher physical capacity, and not, as they assumed, also due to an advantage in cognitive performance. Therefore, the presented study also includes information about physical capacity. As a measure of physical capacity was adopted the average power achieved in the first, second, and third sets of sprint interval training. Because the average power in SIT training was considered by some authors as a parameter correlating with sports performance in mountain bike cycling [15,18]. In the present study, the systematic use of sprint interval training significantly increased the average power output achieved in 30 s repetitions during a sprint interval test, which was accompanied by a decrease in blood pH. During mountain bike cycling races, anaerobic glycolytic metabolism plays an important role, because, during the race, cyclists repeatedly climb uphill [62]. On these climbs, the power often exceeds the value of the maximal aerobic power [17], which results in an increase in the intensity of glycolytic metabolism, consequently an increase in La, and a decrease in blood pH [63]. Therefore, improving the ability to perform an intense exercise with longer (up to the third set of SIT) tolerance of acid-base imbalance may be important for the results of MTB cycling competitions.

## 5. Conclusions

As acute effects of the sprint interval test, the choice reaction time was shortened, and the number of incorrect reactions was reduced. The chronic effects of the polarized training program including sprint interval training, high-intensity interval training, and endurance training showed no change in choice reaction time but reduce the number of incorrect reactions in post-exercise measurements was observed, which was accompanied by an increase in average power output and a decrease in blood pH during the sprint interval test. Such changes are beneficial for achieving good sporting results in mountain bike racing, which is characterized by variable and high intensity, and requires fast correct decision-making.

## Figures and Tables

**Figure 1 ijerph-19-14954-f001:**
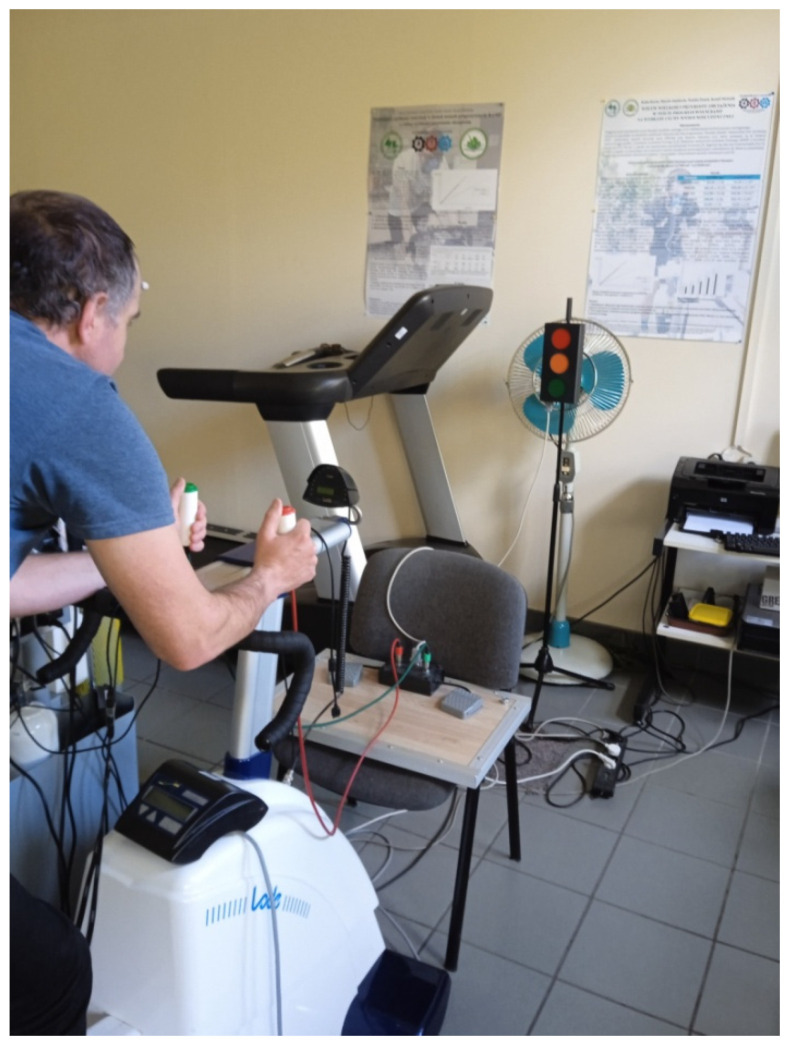
Cyclist’s position during the choice reaction time measurement.

**Table 1 ijerph-19-14954-t001:** Characteristics of the groups of cyclists studied.

Group	Body Height [cm]	Body Weight [kg]	Age [Years]	Pmax [W]	VO_2_max [ml·kg^−1^·min^−1^]
E	175.6 ± 7.1	66.2 ± 10.7	18.7 ± 4.7	331.3 ± 67.4	57.1 ± 6.1
Men in E	178.8 ± 5.0	70.5 ± 9.4	19.5 ± 5.3	368.3 ± 33.3	59.7 ± 4.8
Women in E	167.7 ± 4.5	55.6 ± 5.2	17.2 ± 1.5	210.0 ± 21.3	50.3 ± 2.0
C	174.7 ± 6.9	66.5 ± 9.6	19.6 ± 4.1	338.1 ± 60.3	57.6 ± 7.6
Men in C	176.3 ± 5.7	69.4 ± 8.6	19.9 ± 4.4	360.0 ± 49.1	58.6 ± 8.4
Women in C	169.7 ± 9.3	57.8 ± 7.6	19.0 ± 4.3	272.3 ± 41.2	54.7 ± 3.8

Pmax—maximal aerobic power measured during the incremental test; VO_2_max—maximal oxygen uptake measured during the incremental test; E—experimental group; Men in E—men (*n* = 10) included in the experimental group; Women in E—women (*n* = 4) included in the experimental group; C—control group; Men in C—men (*n* = 9) included in the control group; Women in C—women (*n* = 3) included in the control group; data are presented as mean ± standard deviation.

**Table 2 ijerph-19-14954-t002:** Types of training sessions performed in groups E and C. It is a load during the 1st-3rd week of the experiment, in the following weeks (4th–6th week and 7th–9th week) the training sessions were extended.

Sprint Interval Training (SIT)→ 20 min warm-up → 10 min active rest→ 30 s 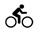 max → 90 s rest → 30 s 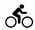 max → 90 s rest → 30 s 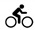 max → 90 s rest → 30 s 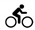 max → 90 s rest → 25 min active rest → 30 s 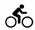 max → 90 s rest → 30 s 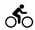 max → 90 s rest → 30 s 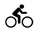 max → 90 s rest → 30 s 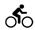 max → 90 s rest → 15 min cooldown
High-intensity interval training (HIIT)→ 20 min warm-up → 10 min active rest→ 5 min 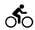 at 85–95% Pmax → 12 min 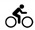 at 50% Pmax → 5 min 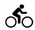 at 85–95% Pmax → 12 min 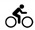 at 50% Pmax → 5 min 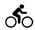 at 85–95% Pmax → 12 min 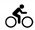 at 50% Pmax → 5 min 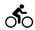 at 85–95% Pmax → 12 min 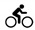 at 50% Pmax → 5 min 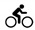 at 85–95% Pmax → 12 min 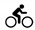 at 50% Pmax → 15 min cooldown
Endurance training (ET)→ 15 min warm-up → 5 min active rest→ 120 min 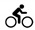 at 55–60% Pmax→ 10 min cooldown

max—the maximal cycling intensity; Pmax—maximal aerobic power measured during the incremental test.

**Table 3 ijerph-19-14954-t003:** Changes in choice reaction time and a number of incorrect reactions measured before and after SITP, as well as before and after the experiment, in groups E and C.

	Experimental Group			Control Group		
	Mean ± SD	95% CI Lower Upper		Mean ± SD	95% CI Lower Upper	
Pre-experiment						
RTav__b_ [s]	0.36 ± 0.04	0.33	0.38	0.35 ± 0.04	0.33	0.38
RTav__e_ [s]	0.34 ± 0.05 ***	0.31	0.37	0.32 ± 0.04 ***	0.30	0.34
RTmin__b_ [s]	0.25 ± 0.03	0.23	0.27	0.24 ± 0.04	0.21	0.26
RTmin__e_ [s]	0.24 ± 0.04	0.22	0.26	0.23 ± 0.04	0.20	0.26
RTmax__b_ [s]	0.53 ± 0.12	0.47	0.60	0.52 ± 0.06	0.48	0.56
RTmax__e_ [s]	0.49 ± 0.13	0.42	0.57	0.45 ± 0.06 ***	0.41	0.48
NIR__b_	3.71 ± 1.33	2.95	4.48	3.83 ± 0.94	3.24	4.43
NIR__e_	2.14 ± 1.03 ***	1.55	2.74	2.75 ± 0.75 ***	2.27	3.23
Post-experiment						
RTav__b_ [s]	0.38 ± 0.06	0.34	0.42	0.33 ± 0.03	0.31	0.35
RTav__e_ [s]	0.34 ± 0.04	0.32	0.37	0.32 ± 0.03	0.30	0.33
RTmin__b_ [s]	0.27 ± 0.05	0.24	0.30	0.24 ± 0.03	0.22	0.25
RTmin__e_ [s]	0.25 ± 0.03	0.23	0.27	0.23 ± 0.03	0.22	0.25
RTmax__b_ [s]	0.57 ± 0.15	0.48	0.65	0.48 ± 0.08	0.43	0.52
RTmax__e_ [s]	0.50 ± 0.09	0.45	0.55	0.46 ± 0.07	0.41	0.51
NIR__b_	2.21 ± 0.97 *	1.65	2.78	3.25 ± 1.21	2.48	4.02
NIR__e_	0.36 ± 0.50 *	0.07	0.64	3.42 ± 1.00 **	2.78	4.05

__b_—baseline measurements taken immediately before a sprint interval testing protocol; __e_—exercise measurements taken immediately after a sprint interval testing protocol; RTav—average choice reaction time; RTmin—minimal/shortest choice reaction time; RTmax—maximal/longest choice reaction time; NIR—number of incorrect reactions; SD—standard deviation; CI—confidence intervals; *—*p* < 0.05—significant difference between pre- and post-experiment values; **—*p* < 0.05—significant difference between experimental and control group; ***—*p* < 0.05—significant difference between baseline__b_ measurements and exercise__e_ measurements.

**Table 4 ijerph-19-14954-t004:** Changes in peak and average anaerobic power achieved during SITP as a result of training programs implemented in groups E and C. Changes were described as chronic effects.

	Experimental Group			Control Group		
	Mean ± SD	95% CI Lower Upper		Mean ± SD	95% CI Lower Upper	
Pre-experiment						
Ppeak_1_ [W]	1185.6 ± 301.0	1011.8	1359.4	1244.9 ± 298.7	1055.1	1434.7
Ppeak_2_ [W]	1044.3 ± 284.2	880.2	1208.4	1102.6 ± 220.2	962.6	1242.5
Ppeak_3_ [W]	1010.5 ± 261.8	859.4	1161.7	1096.0 ± 218.0	957.4	1234.5
Pav_1_ [W]	573.7 ± 111.6	509.3	638.2	584.6 ± 117.1	510.2	658.9
Pav_2_ [W]	575.0 ± 108.9	512.1	637.9	600.8 ± 119.3	525.0	676.6
Pav_3_ [W]	564.3 ± 108.3	501.7	626.8	588.4 ± 118.2	513.3	663.5
Post-experiment						
Ppeak_1_ [W]	1143.2 ± 262.7	991.5	1294.9	1199.1 ± 298.0	1009.8	1388.5
Ppeak_2_ [W]	1096.0 ± 240.4	957.2	1234.8	1098.8 ± 242.7	944.6	1253.0
Ppeak_3_ [W]	1086.4 ± 249.7	935.5	1237.3	1094.2 ± 266.1	925.2	1263.3
Pav_1_ [W]	595.0 ± 111.2 *	530.8	659.2	600.6 ± 112.0	529.5	671.8
Pav_2_ [W]	594.5 ± 109.2	531.4	657.5	604.4 ± 120.3	528.0	680.8
Pav_3_ [W]	592.2 ± 109.4 *	526.1	658.3	591.4 ± 119.7	515.3	667.4

Ppeak_1_—peak power achieved during the first sets (repetitions 1–4) of the sprint interval testing protocol; Ppeak_2_—peak power achieved during the second sets (repetitions 5–8) of the sprint interval testing protocol; Ppeak_3_—peak power achieved during the third sets (repetitions 9–12) of the sprint interval testing protocol; Pav_1_—average power calculated from four repetitions of the first sets of the sprint interval testing protocol; Pav_2_—average power calculated from four repetitions of the second sets of the sprint interval testing protocol; Pav_3_—average power calculated from four repetitions of the third sets of the sprint interval testing protocol; SD—standard deviation; CI—confidence intervals; *—*p* < 0.05—significant difference between pre- and post-experiment value.

**Table 5 ijerph-19-14954-t005:** Biochemical and physiological changes during SITP as a result of training programs implemented in groups E and C. Changes were described as chronic effects.

	Experimental Group			Control Group		
	Mean ± SD	95% CI Lower Upper		Mean ± SD	95% CI Lower Upper	
Pre-experiment						
La_1_ [mmol/L]	17.19 ± 2.47	15.77	18.61	18.25 ± 2.57	16.62	19.88
La_2_ [mmol/L]	17.74 ± 2.22	16.46	19.03	18.14 ± 2.42	16.60	19.68
La_3_ [mmol/L]	16.23 ± 2.68	14.68	17.77	18.08 ± 2.23	16.67	19.50
pH_1_	7.07 ± 0.05	7.04	7.11	7.04 ± 0.05	7.01	7.08
pH_2_	7.10 ± 0.04	7.08	7.13	7.06 ± 0.05	7.02	7.09
pH_3_	7.13 ± 0.05	7.10	7.16	7.06 ± 0.07	7.02	7.11
HR_1_ [b/min]	181.07 ± 5.93	177.65	184.50	179.42 ± 4.36	176.65	182.19
HR_2_ [b/min]	179.21 ± 5.16	176.23	182.20	178.58 ± 4.81	175.53	181.64
HR_3_ [b/min]	178.29 ± 5.57	175.07	181.50	177.08 ± 5.68	173.47	180.69
Post-experiment						
La_1_ [mmol/L]	16.57 ± 2.17	15.32	17.82	18.15 ± 2.72	16.42	19.88
La_2_ [mmol/L]	16.57 ± 1.65	15.62	17.52	17.63 ± 2.84	15.82	19.43
La_3_ [mmol/L]	16.34 ± 2.67	14.80	17.88	17.66 ± 3.47	15.45	19.86
pH_1_	7.09 ± 0.06	7.06	7.12	7.08 ± 0.06 *	7.04	7.12
pH_2_	7.09 ± 0.04	7.07	7.12	7.08 ± 0.07	7.04	7.13
pH_3_	7.10 ± 0.04 *	7.07	7.12	7.09 ± 0.08 *	7.05	7.14
HR_1_ [b/min]	181.64 ± 5.73	178.33	185.0	179.42 ± 4.38	176.63	182.20
HR_2_ [b/min]	180.21 ± 5.41	177.09	183.34	177.92 ± 3.99	175.38	180.45
HR_3_ [b/min]	177.14 ± 4.97	174.27	180.02	176.17 ± 4.39	173.38	178.95

_1_—measurement taken after the first sets (repetitions 1–4) of the sprint interval testing protocol; _2_—measurement taken after the second sets (repetitions 5–8) of the sprint interval testing protocol; _3_—measurement taken after the third sets (repetitions 9–12) of the sprint interval testing protocol; La—blood lactate concentration; pH—blood pH value; HR—peak heart rate value; SD—standard deviation; CI—confidence intervals; *—*p* < 0.05—significant difference between pre- and post-experiment value.

## Data Availability

The data presented in this study are available on request from the corresponding author.

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
