# Peer review of "Acute Effects of Sprint Interval Training and Chronic Effects of Polarized Training (Sprint Interval Training, High Intensity Interval Training, and Endurance Training) on Choice Reaction Time in Mountain Bike Cyclists"

_ijerph, 2022, doi:10.3390/ijerph192214954_

Round 1

Reviewer 1 Report (New Reviewer)

I am grateful for the opportunity to review this manuscript titled "Acute and chronic effects of sprint interval training on choice reaction time in mountain bike cyclists”. The purpose of this study was to evaluate the acute and chronic effects of sprint interval training on choice reaction time in cyclists. The data collected in this study may affirm or expand on available literature.

This study is of interest to the IJERPH readers and seems to provide some new findings, applicable to the fields of training. However, the points mentioned in the “Specific comments” section below should be considered and the manuscript amended accordingly before being considered for publication. Also, a native English speaker should read and review the manuscript before its submission.

Specific comments

Abstract

1. The authors could briefly describe the experimental protocols.

2. It would be appropriate for authors to introduce statistical values in the abstract (i.e., p-value, effect size, ...).

Introduction

3. The introduction is well written, but the authors need to highlight in the introduction the contribution of their work to the area.

4. The study hypothesis is missing.

Materials and Methods

2.2. Test procedures

5. Were the same researchers who performed the assessments pre and post? In addition, it is not mentioned if the experimenters were blinded to the research question.

2.2.2. Sprint interval testing protocol (SITP)

6. Why did the authors select a rest time of 25 min between sets? One of the main characteristics of the SITP is the short time needed for each session. But with this configuration, the sessions take more than 2 hours. It is neither realistic nor applicable. I would like to know the opinion of the authors on this point.

2.4. Statistical analysis

7. Please rewrite this sentence. It is confusing.

The Student's t-test was used to determine whether there were statistical differences in reaction time parameters when tested before and after the sprint interval testing protocol that was performed before the experiment.

8. Authors should describe Æž2.

Results

9. Line 252: Change “K” to “C”.

10. I don't think this is the most appropriate description to describe table 5.

Author Response

Reviewer 1

I am grateful for the opportunity to review this manuscript titled "Acute and chronic effects of sprint interval training on choice reaction time in mountain bike cyclists”. The purpose of this study was to evaluate the acute and chronic effects of sprint interval training on choice reaction time in cyclists. The data collected in this study may affirm or expand on available literature.

This study is of interest to the IJERPH readers and seems to provide some new findings, applicable to the fields of training. However, the points mentioned in the “Specific comments” section below should be considered and the manuscript amended accordingly before being considered for publication. Also, a native English speaker should read and review the manuscript before its submission.

- Thank you very much for your review and the valuable comments.

Specific comments

Abstract

  1. The authors could briefly describe the experimental protocols.

- Indeed, the description was quite sparse, unfortunately often the limitation of our abstracts is due to the limit of words recommended by the Journal.

In the current version of the manuscript the experimental protocols are described in more detail.

  1. It would be appropriate for authors to introduce statistical values in the abstract (i.e., p-value, effect size, ...).

- Statistical values were added in the abstract, for p-value and effect size.

Introduction

  1. The introduction is well written, but the authors need to highlight in the introduction the contribution of their work to the area.

- As suggested, some information has been added to the Introduction section.

  1. The study hypothesis is missing.

- Thank you for catching this error, the hypothesis has been added.

Materials and Methods

2.2. Test procedures

  1. Were the same researchers who performed the assessments pre and post? In addition, it is not mentioned if the experimenters were blinded to the research question.

- Yes, The same two researchers performed the measurement both before and after the experiment in all participants.

Yes, the researchers were blind to the research question. These are laboratory employees who are well versed in the devices and their operation and are not interested in the purpose of the research.

This information was added in the Materials and Methods section, in point 2.2

2.2.2. Sprint interval testing protocol (SITP)

  1. Why did the authors select a rest time of 25 min between sets? One of the main characteristics of the SITP is the short time needed for each session. But with this configuration, the sessions take more than 2 hours. It is neither realistic nor applicable. I would like to know the opinion of the authors on this point.

- We agree with the Reviewer that the current trend is to keep the interval training sessions short. However, in mountain bike cycling we do not care about short training sessions, on the contrary. Competition during the mountain bike race in the age categories over 19 and over 23, lasts about 1 hour and 30 minutes, plus an intense warm-up. So we deliberately try to extend this training over time so that the body learns to repeat intense effort for a long time.

For several years, in our research, we have been dealing, among other things, with determining the appropriate break time between the sets of maximal repetitions in SIT training. The time of the active break in the SIT training was selected individually for each cyclist. First of all, we wanted to achieve a level close to the acid-base balance during the break between sets. The second factor that we took into account was the amount of work done and the power achieved, we tried to ensure that these values ​​were similar in subsequent sets, which was possible with the use of a long rest break.

In our previous articles, we explained how we determine the optimal break time between sets for mountain bike cyclists. Based on these studies, we have determined time in the range of 22-27 minutes, in the presented study we chose an average of 25 minutes.

  1. Hebisz P, Hebisz R, BakoÅ„ska-PacoÅ„ E, ZatoÅ„ M. Acute hematological response to a single dose of sprint interval training in competitive cyclists. Science & Sports  2017; 32(6): 369-375:  “The sets were separated by 20–40 min active recovery performed at moderate intensity (70–75% HRmax), with recovery duration tailored to each participant by the time needed to return to relative acid–base homeostasis. Blood pH was measured using a RAPIDLab 348 blood gas system (Siemens Healthcare, Germany) 20 min into the cool-down and then every 5 min until attaining a level of 7.35. This value was adopted as normal as it was assumed baseline pH was unattainable due to the active nature of the cool-down.
  2. Hebisz R, Hebisz P, ZatoÅ„ M. Work efficiency in repeated sets of sprint interval exercise in cyclists. The Journal of Sports Medicine and Physical Fitness 2017; 57(3): 195-201:After finishing a set the participant again performed an active cool-down by first pedaling with no external load for 2 min which was then increased 2 W/kg. The duration of the cool-down was dependent on the restoration of acid–base balance and lasted approximately 25–40 min. Arterialized capillary blood pH was measured 25 min into the cool-down and then every 5 min until reaching 7.35 pH. This level was adopted as normal as it was assumed baseline pH was unattainable due to the active nature of the cool-down. After active recovery was completed, the participant performed a low-intensity bout (0.5 W/kg) before starting the next set.

- Brief information on determining the time between sets has been added in the Materials and Methods section.

2.4. Statistical analysis

  1. Please rewrite this sentence. It is confusing.

The Student's t-test was used to determine whether there were statistical differences in reaction time parameters when tested before and after the sprint interval testing protocol that was performed before the experiment.

- With reference to the comment of the second Reviewer and the change in the statistical analysis, this sentence has been deleted.

  1. Authors should describe Æž2.

- As suggested, the eta squared (η2) is described in the indicated section of the manuscript.

Results

  1. Line 252: Change “K” to “C”.

- Done

  1. I don't think this is the most appropriate description to describe table 5.

- The description of Table 5 has been changed. Similarly, the description of Table 4 was also changed. As a new table was added, the numbering of the tables has changed:

Table 4 → Table 5

Table 5 → Table 6

Reviewer 2 Report (New Reviewer)

Thanks for your invitation to review this study. However, there is a major concern of the study. The main measurement of choice reaction time is not supported by the physiological/psychological evidence. It is not convincible to us the association of SITP performance and blood pH measures. I suggest the authors justified the demand to evaluate the choice rection time for mountain bike cyclists. The decision making during the mountain bike race are different to the visual reaction time (cognitive process). Although the authors review the literature regarding the polarized training intervention and reaction time performance, no statement to justify the rationale to evaluate the choice reaction time during the 9 weeks sprint interval training (no attention/visual reaction to stimuli).

Abstract

1.        Information for the experimental design and measuring variables are unclear.

2.        The results are not fully presented here.

3.        The conclusion should be rephrased and rewritten

Introduction

1.     The investigation for acute and chronic effects of the 9 weeks sprint interval training is unclear.

2.     The rationale to conduct a 9 weeks sprint interval training intervention for change of choice reaction time is absent.

Methods

1.        How the authors assigned the participants to the groups.

2.        Please provide the sample size estimation and power report.

3.        It is better to tabulate the training program of the study (line 201-229).

Results

1.     Are the results normally distributed?

2.     It is tricky to see the authors present Cohen’s D and eta square in one study.

3.     The presentation from Table 4-6 can be improved by change the layout of group presentation.

4.     The correlations are not performed appropriately. Two groups should be pooled together for comparison since the variance of training intervention. Additionally, the blood lactate concentration and pH were pooled with the difference between set 1 and set 3 but reaction time was before and after the SITP test.

l   Table 2 and Figure 1, 3 is abundant.

l   There are a number of typos and grammatical errors in the manuscript. It should be edited and proofreading before next round of review.  

Author Response

Reviewer 2

Thanks for your invitation to review this study. However, there is a major concern of the study. The main measurement of choice reaction time is not supported by the physiological/psychological evidence. It is not convincible to us the association of SITP performance and blood pH measures. I suggest the authors justified the demand to evaluate the choice rection time for mountain bike cyclists. The decision making during the mountain bike race are different to the visual reaction time (cognitive process). Although the authors review the literature regarding the polarized training intervention and reaction time performance, no statement to justify the rationale to evaluate the choice reaction time during the 9 weeks sprint interval training (no attention/visual reaction to stimuli).

- Thank you very much for your review and the valuable comments.

Abstract

  1. Information for the experimental design and measuring variables are unclear.

- Indeed, the description was unclear, unfortunately often the limitation of our abstracts is due to the limit of words recommended by the Journal.

In the current version of the manuscript the experimental design and measuring variables are described in more detail.

  1. The results are not fully presented here.

- The description of the results has been improved.

  1. The conclusion should be rephrased and rewritten

        - As suggested, the conclusions were rewritten.

Introduction

  1. The investigation for acute and chronic effects of the 9 weeks sprint interval training is unclear.

- In the Introduction section, we have added some information relating to the acute and chronic effects of training.

  1. The rationale to conduct a 9 weeks sprint interval training intervention for change of choice reaction time is absent.

      - As suggested, in the current version of the manuscript we referred to planning a 9 weeks sprint interval training intervention to change the choice reaction time.

Methods

  1. How the authors assigned the participants to the groups.

- The participants were randomly assigned to the groups. Such information was added in the Materials and Methods section.

  1. Please provide the sample size estimation and power report.

        - Indeed, there was no such information, it was corrected.

Prior to the experiment, using the G-Power 3.1.9.4 software, we estimated that the minimum total sample size for Anova with repeated measurements is 16 people, assuming that we expect a strong effect size, i.e. a partial η2 ≥ 0.14 at p <0.05 (Lakens 2013).

Such information was added at the end of the Materials and Methods section.

  1. It is better to tabulate the training program of the study (line 201-229).

        - As suggested, the training program of the study was additionally tabulated (Table 3). The table has been added to the Materials and Methods section.

Results

  1. Are the results normally distributed?

- The similarity to the normal distribution was calculated using the Kolmogorov-Smirnov test. The distribution of the analyzed parameters was close to the normal distribution. Relevant information has been added in the Material and Methods, and Results sections.

  1. It is tricky to see the authors present Cohen’s D and eta square in one study.

      - We abandoned the use of Student's t-test to compare changes in reaction time in the measurements before and after the sprint interval testing protocol. Therefore, we also do not present Cohen's D value for showing the size effect. In the current version of the manuscript, we used an analysis of variance with repeated measures to identify statistically significant differences in  reaction time measured before and after the sprint interval testing protocol. Therefore, the size effect was presented using eta square (η2), similarly to the other analyzes performed in these studies.

This information has been added in the Materials and Methods, and Results section.

  1. The presentation from Table 4-6 can be improved by change the layout of group presentation.

- I'm very sorry, but I don't understand this comment. I have no idea how to change the layout.

  1. The correlations are not performed appropriately. Two groups should be pooled together for comparison since the variance of training intervention. Additionally, the blood lactate concentration and pH were pooled with the difference between set 1 and set 3 but reaction time was before and after the SITP test.

      - As suggested by the Reviewer, we made correlation for the two groups pooled together, before and after the experiment. Such an analysis did not allow to obtain statistically significant correlations. We obtained r≈ 0.26 at p greater than 0.05 for 52 measurements. Therefore, we did not publish the results on the correlation between reaction time and pH, and the correlation between reaction time and blood lactate concentration.

- These correlations have been removed from the Results section. In addition, the text in the Discussion section referring to these correlations has been removed.

l   Table 2 and Figure 1, 3 is abundant.

  • Sorry to write in a similar tone again, but I don't quite understand the suggestion. Does that mean Table 2 and Figure 1,3 are unnecessary and should I delete them. If so, of course I can delete them.

l   There are a number of typos and grammatical errors in the manuscript. It should be edited and proofreading before next round of review. 

- The revised manuscript was sent to the native speaker. However, our permanent translator is on vacation, so a new person checked the revised version of the manuscript.

Round 2

Reviewer 1 Report (New Reviewer)

The authors have improved the manuscript according to the recommendations and reaching the minimum standards.

Author Response

Reviewer 1

The authors have improved the manuscript according to the recommendations and reaching the minimum standards.

- Thank you very much for your review and positive feedback.

Reviewer 2 Report (New Reviewer)

1.        Thanks for your revision work. I have a question about the title of the study. The authors stated the polarized training programme for the cyclists rather than sprint interval training along.

2.        In term of the polarized training programme, as questioning in my last review, justification to see training benefits for choice reaction is relatively weak. There is no link between the cycling training and chronic time reaction in the revision since you have no metabolic/cardiovascular measures at the cerebral level.

3.        The participants include 10 men and 4 women in the training group, while 9 men and 3 women were in the control group. It is better to show the physical characteristics in men and women.

4.        I don’t think it is appropriate to pool all data for the comparisons in Table 4. Investigation for acute effect using one group but for chronic effects using two groups?

5.        Please avoiding to laundry the review in the discussion section. The authors are encouraged to interpret the findings concisely. For example, how the training group improve the incorrect reactions after the training period, but control group increase post-exercise incorrect reactions after the training period? How are the findings associated with cycling performance and blood pH?

 The presentation from Table 4-6 can be improved by change the layout of group presentation. I'm very sorry, but I don't understand this comment. I have no idea how to change the layout.

The presentation of the tables should focus on the group comparison.

Table 2 and Figure 1, 3 is abundant. Sorry to write in a similar tone again, but I don't quite understand the suggestion. Does that mean Table 2 and Figure 1,3 are unnecessary and should I delete them. If so, of course I can delete them.

Please remove the table 2 and figures 1 and 3 in the first version of manuscript.

Author Response

Reviewer 2

- Thank you very much for your review and the valuable comments, they have greatly contributed to our paper.

  1. Thanks for your revision work. I have a question about the title of the study. The authors stated the polarized training programme for the cyclists rather than sprint interval training along.

        - Indeed, the title was inappropriate.

It has been corrected to: "Acute effects of sprint interval training and chronic effects of polarized training (sprint interval training, high intensity interval training, and endurance training) on choice reaction time in mountain bike cyclists"

Throughout the manuscript, terms such as "...acute and chronic effects of sprint interval training ..." have been corrected analogously.

  1. In term of the polarized training programme, as questioning in my last review, justification to see training benefits for choice reaction is relatively weak. There is no link between the cycling training and chronic time reaction in the revision since you have no metabolic/cardiovascular measures at the cerebral level.

        - At the end of the Introduction section, before the study aim, we have added information about the molecular, metabolic, and cardiovascular changes that can occur at the cerebral level as a result of polarized training or high-intensity training. We hope that this text well justifies the need to investigate the effect of a polarized training program on changes in reaction time.

  1. The participants include 10 men and 4 women in the training group, while 9 men and 3 women were in the control group. It is better to show the physical characteristics in men and women.

        - In Table 1, we added the characteristics of men and women.

  1. I don’t think it is appropriate to pool all data for the comparisons in Table 4. Investigation for acute effect using one group but for chronic effects using two groups?

        - As suggested, we performed an acute effect analysis divided into experimental and control group. Therefore, Table 4 became unnecessary and was deleted. The results of the analysis (Scheffe post-hoc test) are presented in the next Table 5 and in the additional description in the Materials and Methods section.

As some tables have been deleted, the table numbering has changed:

Table 5 → Table 3

Table 6 → Table 4

Table 7 → Table 5

  1. Please avoiding to laundry the review in the discussion section. The authors are encouraged to interpret the findings concisely. For example, how the training group improve the incorrect reactions after the training period, but control group increase post-exercise incorrect reactions after the training period? How are the findings associated with cycling performance and blood pH?

        - In the Discussion section, we removed information that was of little relevance to the explanation of the observed results.

In addition, we have added information about chronic changes as a result of the training period, in relation to: the number of incorrect reactions, cycling performance and blood pH.

 The presentation from Table 4-6 can be improved by change the layout of group presentation. I'm very sorry, but I don't understand this comment. I have no idea how to change the layout.

The presentation of the tables should focus on the group comparison.

  • The layout of the tables has been changed to focus on the group comparison.

Table 2 and Figure 1, 3 is abundant. Sorry to write in a similar tone again, but I don't quite understand the suggestion. Does that mean Table 2 and Figure 1,3 are unnecessary and should I delete them. If so, of course I can delete them.

Please remove the table 2 and figures 1 and 3 in the first version of manuscript.

  • The table and figures have been removed.

This manuscript is a resubmission of an earlier submission. The following is a list of the peer review reports and author responses from that submission.

Round 1

Reviewer 1 Report

Major questions

- Pmax: the maximum mechanical power obtained during the IT is being called maximum aerobic power (line 120). This is not the same thing: aerobic power involves calculating the energy equivalent for oxygen consumption and its value is not the same as mechanical power. In this sense, I suggest not to use Pmax maximum aerobic power denomination, but maximum mechanical power, because from the way it is described it seems that the metabolic conversion was carried out (change throughout the text). (Peyré-Tartaruga and Coertjens, 2018: https://www.frontiersin.org/articles/10.3389/fphys.2018.01789/full ).

Minor questions

- Line 97: ... (?)

- Table 01: Please, correct: “ml-1·min-1·kg” (ml*kg-1*min-1)

- Line 110: ... (?)

- 137: “Repetitions were divided into sets and 4 repetitions were performed in each set”. Suggestion: ...into 3 sets...

- Line 533: ... Melabolic and hormonal responses to exercise in children and adolescents.

Author Response

Reviewer 1

- Thank you very much for your review and the valuable comments.

Major questions

- Pmax: the maximum mechanical power obtained during the IT is being called maximum aerobic power (line 120). This is not the same thing: aerobic power involves calculating the energy equivalent for oxygen consumption and its value is not the same as mechanical power. In this sense, I suggest not to use Pmax maximum aerobic power denomination, but maximum mechanical power, because from the way it is described it seems that the metabolic conversion was carried out (change throughout the text). (Peyré-Tartaruga and Coertjens, 2018: https://www.frontiersin.org/articles/10.3389/fphys.2018.01789/full ).

- Thank you for the useful comment, the name has been corrected throughout the text of the article.

Minor questions

- Line 97: ... (?)

- The name of the University has been completed.

- Table 01: Please, correct: “ml-1·min-1·kg” (ml*kg-1*min-1)

- Done

- Line 110: ... (?)

- The name of the University has been completed.

- 137: “Repetitions were divided into sets and 4 repetitions were performed in each set”. Suggestion: ...into 3 sets...

- The sentence has been corrected.

- Line 533: ... Melabolic and hormonal responses to exercise in children and adolescents.

- Thank you for catching this mistake. This has been corrected.

Reviewer 2 Report

Line 62: do these measures correlate with race performance. There are references in place. If not speculate why they do not.

Line 66: Please replace with “fatigue increases RT among athletes…”

Line 74: maybe worth highlighting the differences in the previous exercise training protocols to maybe speculate why there were different responses observed.

Line 77: I was under the impression that a polarized training model consists of spending ~80% of training time below ventilatory threshold 1, indicating light intensity, while ~20 percent is spent above ventilatory threshold 2, indicating high intensity. Training between VT1 and VT2 is avoided, indicating moderate intensities. Can you clarify this?

Line 81: Why choice RT? Does this correlate with mountain bike performance. Why not just simple RT?

Line 119-120: Can you provide a reference for this? Typically if a stage is not completed the powerout of final completed stage is added to (time completed of failed stage/increment of stages).

Line 121: Was verification protocol used to verify VO2max and Pmax?

Line 128: Was VO2max determined as the highest 30-second average or was it the average of the final two 30-second samples. Just clarification here please.

Line 131: Was SITP performed on a separate day from IT? If so how many days between?

Line 136: Was the powerout rpm-dependent for the SITP?

Line 216: Was there any matching for training stress or work completed between the two groups? I know a SIT session was substituted with HIIT in the control group but there should still be a control for training time intensity.

Line 316: Discussion should begin with a paragraph reiterating the research question, hypothesis, and the findings before diving into discussing findings.

Line 355: maybe worth confirming when the reaction time assessments were conducted after the training session. Within 10 seconds or exercise, 1 minute or 5 minutes. The deleterious effects are often reduced the further out from exercise so timing is important and worth discussing.

Author Response

Reviewer 2

- Thank you very much for your review and the valuable comments.

Line 62: do these measures correlate with race performance. There are references in place. If not speculate why they do not.

- Yes, these measurements are correlated with race performance. The sentence has been corrected.

Line 66: Please replace with “fatigue increases RT among athletes…”

- The sentence has been corrected. 

Line 74: maybe worth highlighting the differences in the previous exercise training protocols to maybe speculate why there were different responses observed.

- The differences in training protocols and study participants were highlighted.

Line 77: I was under the impression that a polarized training model consists of spending ~80% of training time below ventilatory threshold 1, indicating light intensity, while ~20 percent is spent above ventilatory threshold 2, indicating high intensity. Training between VT1 and VT2 is avoided, indicating moderate intensities. Can you clarify this?

 - Yes, that's right. Training polarization consists in creating training programs that include efforts of different intensity - low at the level of the first ventilatory threshold (VT1) and high above the second ventilatory threshold (VT2) [25-27]. The volume of low-intensity training sessions is approximately 80% of the total training volume, while the high-intensity training is approximately 20% of the total training volume (Zapata-Lamana et al. 2018). In polarized training programs, moderate-intensity training at VT2 is not used [25,28,29], or these training sessions account for a small part of the training load (up to 5% of the total training volume) (Röhrken et al. 2020). In the studies by Hebisz et al. [28, 29], polarized training was defined as training including SIT, HIIT and LIT trainings, with the exclusion of training at intensity of the VT2. These studies [28, 29] have shown that polarized training is effective in the development of aerobic capacity assessed by VO2max and Pmax.

With reference to this comment, the polarized training program is described in more detail in the indicated part of the manuscript.                 

Line 81: Why choice RT? Does this correlate with mountain bike performance. Why not just simple RT?

- In the available literature, we have not found any article that would show the correlation of choice reaction time with performance in mountain bike cycling. However, based on the literature we have read relating to the measurement of athletes' reaction time in other sport disciplines, we have chosen a choice reaction time.

We had in mind the characteristics of mountain bike cycling, as well as the nature of the decisions and reactions made by mountain bike cyclists during races. Mountain biking races take place on narrow forest paths with uneven ground, sharp turns, obstacles in the form of stones and roots, in changing weather conditions and in direct contact with other cyclists whose behavior cannot be predicted.

Line 119-120: Can you provide a reference for this? Typically if a stage is not completed the powerout of final completed stage is added to (time completed of failed stage/increment of stages).

- References have been added to the indicated text presenting the method of calculating the maximal power.

Line 121: Was verification protocol used to verify VO2max and Pmax?

 - The test protocol with VO2max and Pmax verification was not used in the carried out research. The plateau phase of the maximal oxygen uptake was treated as a confirmation of the VO2max obtaining. Participants who did not reach the plateau phase of VO2max were not included in the data analysis. Therefore, the manuscript and the description of the results included 26 cyclists out of the 30 who started the study. However, it is difficult not to agree with the Reviewer’s suggestion that such verification should be carried out. We will take this suggestion into account in our next research.

Brief information about this has been added in the Materials and Methods section.

Line 128: Was VO2max determined as the highest 30-second average or was it the average of the final two 30-second samples. Just clarification here please.

- VO2max determined as the highest 30-second average. This information has been added in the indicated fragment of the Materials and Methods section.

Line 131: Was SITP performed on a separate day from IT? If so how many days between?

- Indeed, this information was missing, our mistake. Yes, the tests were performed on separate days. There was a break of 48 hours between IT and SITP. This information was added to the Materials and Methods section in point 2.2. Test procedures.

Line 136: Was the power out rpm-dependent for the SITP?

- No, the power output was not rpm dependent in the SITP.

Line 216: Was there any matching for training stress or work completed between the two groups? I know a SIT session was substituted with HIIT in the control group but there should still be a control for training time intensity.

 - The training program used in the control group has been described in more detail.

Line 316: Discussion should begin with a paragraph reiterating the research question, hypothesis, and the findings before diving into discussing findings.

- As suggested, the beginning of the Discussion section has been changed.

Line 355: maybe worth confirming when the reaction time assessments were conducted after the training session. Within 10 seconds or exercise, 1 minute or 5 minutes. The deleterious effects are often reduced the further out from exercise so timing is important and worth discussing.

- As suggested, the following text has been added to the Discussion section:

It is possible that the differences between the effects in the studies in question are influenced by the timing of the RT measurement. In the studies by Pavelka et al. [24] and Gierczuk et al. [10], similarly to the presented study, the measurement of RT was performed immediately after the end of exercise. However, the duration of the measurements was different: on average 4 to 5 minutes - depending on the speed of each participant [24], the 30 seconds at each participant [10], while in the presented study it was 120 seconds at each participant. It is worth emphasizing that the deleterious effects are often reduced the further out from exercise so the appropriate  measuring time is important [6,7].

Round 2

Reviewer 2 Report

The authors should be commended for addressing comments.

One additional comment needs to be addressed. In your responses you mentioned that 30 participants began the study but only 26 were included in analysis based on the obtainment of a plateau in VO2. This needs to be mentioned in the results section as well as the criteria for inclusion based on obtainment of a plateau in VO2 in the methods.

Author Response

Reviewer 2

One additional comment needs to be addressed. In your responses you mentioned that 30 participants began the study but only 26 were included in analysis based on the obtainment of a plateau in VO2. This needs to be mentioned in the results section as well as the criteria for inclusion based on obtainment of a plateau in VO2 in the methods.

- As suggested, information has been added that 30 participants started the study, but only 26 were included in the analysis, both in the Methods and Results section.

Methods section: Participants who did not reach the plateau phase of VO2max were not included in the data analysis. Therefore, in the presented study, 26 cyclists out of 30 who started the study were included in the data analysis and description of the results.

Results section: The presented study originally included 30 cyclists, but 4 cyclists did not reach the plateau phase of VO2max, which was the criterion for inclusion in the data analysis. So the results presented in this section include 26 cyclists.

Round 3

Reviewer 2 Report

Thank you for addressing my comments